# Optical Properties and Interference Effects of the Lens Mitochondrion

**DOI:** 10.3390/membranes13060610

**Published:** 2023-06-20

**Authors:** Felix Margadant, Zakaria Almsherqi, Xiaochun Xu, Yuru Deng

**Affiliations:** 1Department of Molecular Cell Biology, Institute for Cancer Research, Radium University Hospital, 0379 Oslo, Norway; 2Department of Physiology, National University of Singapore, Singapore 117593, Singapore; phszama@nus.edu.sg; 3Mechanobiology Institute, National University of Singapore, Singapore 117411, Singapore; 4Wenzhou Institute, University of Chinese Academy of Sciences, Wenzhou 325001, China

**Keywords:** lens mitochondria, cubic membranes, tree shrew, quasi-bandgap, Monte Carlo method, Hecht’s solution

## Abstract

The lens mitochondrion of the tree shrew, located along the optical pathway between the lens and photoreceptors, has been investigated. The results suggest that the lens mitochondrion acts as a quasi-bandgap or imperfect photonic crystal. Interference effects cause a shift in the focus and introduce wavelength-dependent behavior similar to dispersion. Optical channels within the mitochondrion form a mild waveguide, preferentially propagating light within certain compartments. The lens mitochondrion also functions as an imperfect UV-shielding interference filter. Overall, this study provides insights into the dual role of the lens mitochondrion and the complex behavior of light within biological systems.

## 1. Introduction

Mitochondria are organelles with a complex structure that consists of two membranes: the outer membrane and the inner membrane. The shape and organization of the inner membrane, called cristae, have traditionally been considered uniform in shape and size. However, recent studies using advanced imaging techniques have revealed that the structure of mitochondria is highly dynamic [1], with variations in the inner mitochondrial membrane ranging from lamellar, tubular, vesicular, zigzag, or reticular net forms to highly convoluted cubic membrane structures [2]. Studies of tree shrew species have found larger-sized mitochondria in the retinal cones, called “lens” mitochondria, with a distinct pattern of multi-layered cristae (Figure 1A). These larger-sized mitochondria, measuring 2–8 µm in diameter, have been found in the retinal cones of the common tree shrew (*Tupaia Glis*) [3] and the northern tree shrew (*Tupaia Belangeri*) [4]. Despite the discovery that the true 3D structure of mitochondria is a multi-layered cubic membrane of the gyroid subtype consisting of twelve layers organized in six pairs [5] (Figure 1B), the potential function in the optical path is still illusive. From a functional perspective, the retinas of certain tree shrew species are unique among mammals, with up to 95% cone cells and only a few scattered rods [3]. The all-cone retina, with less than 5% rods, raises concerns about its function. The maximum diameter of the inner segments of the cones in Tupaia is much wider than the diameter of the outer segments, resulting in widely spaced outer segments rather than densely packed outer segments. In turn, this may lead to a significant percentage of the light penetrating the inner segment being absorbed by melanin pigments instead of being processed by the receptors in the outer segment if the light is not refracted strongly toward the central longitudinal axis of the outer segment. Additionally, the limited number of rods in Tupaia’s retina compared to higher primates may shift the wavelength limit of the visual response toward the biologically damaging UV region. Due to these unsatisfactory circumstances, it has been suggested that the distinctive size, dense matrix, and specialized multi-lamellar system of cristae of Tupaia’s highly refractive mitochondria may have additional optical roles that maximize the all-cone retina function.

In this study, we evaluate the potential optical characteristics and suggest new functions for lens mitochondria. We propose that if the “lens” mitochondria exhibited sufficient rejection of short-wavelength light, it could then have a dual purpose of serving as a ball lens to focus the light onto the receptor as well as filtering out high-energy photons and hence protecting the receptor from photo-dissociation.

We devise a Monte Carlo method in which a finite but large number of rays with the same frequency, polarization, phase, and direction are fired from a 2D grid in the yz plane to simulate the energy transport and polarization of a plane lightwave through the lens mitochondrion of a tree shrew. Hecht’s solution is used to solve the transition of the light through the stratified layers of the membrane, treating each layer as an element that changes the phase, direction of propagation, and fractions of propagated and reflected energy, and it describes this transition using a 2 × 2 matrix.

Our novel approach aims to investigate the optical properties and interference effects of the lens mitochondrion in tree shrews. In comparison to the commonly used finite difference time domain (FDTD) methods, our method offers incremental novelty by significantly reducing computational time, memory requirements, and complexity. Moreover, our simulation employs a coarser mesh, albeit with a trade-off in computational accuracy. Notably, our method surpasses pure ray-tracing approaches by accounting for diffraction and local interference phenomena. This feature is a crucial aspect that distinguishes our study from previous research as it enables a more comprehensive understanding of the behavior of light within the lens mitochondrion. The reduced computational resources and the incorporation of diffraction and local interference phenomena provide a more realistic representation of light transport in biological systems.

These innovative aspects of our methodology open up avenues for further exploration and applications in the field of optical device design. The findings contribute to the advancement of knowledge about the unique optical characteristics of biological systems and underscore the potential for utilizing such insights in the development of efficient and accurate optical simulations.

## 2. Materials and Methods

Unlike extended cubic phases of comparable structure size, the lens mitochondrion (Figure 1A) is confined within a minute sphere (Figure 1B) with a radius of approximately 1 µm [3,4,5]. Consequently, predicting its optical properties becomes challenging. The refractive index variations between the compartments are not sufficiently steep to generate significant interference effects. Furthermore, it remains uncertain whether the relatively thick separation of the compartments, composed of 12 distinct films of lipid bilayers, serves any optical purpose.

### 2.1. Simulation Approach

Successful approaches for simulating the properties of extended photonic crystals involve solving the light transport as a plane wave in each spatial direction and then combining the probabilities for each spatial direction to obtain the overall superposition [6]. However, this concept is only applicable when the object is much larger than the wavelength and exhibits strong, large-scale regularity, which is not the case for the lens mitochondrion.

Another computationally expensive approach is to solve the light transport for a polarized collimated source attached to a unit cell of interest and calculate the transport across an assembly of unit cells by integrating over their surfaces rather than their volume. While this approach is suitable for finite-sized unit cell assemblies, it cannot be directly applied to the spherical boundary of the mitochondrion due to the large fraction of partial unit cells. For instance, if we consider a diameter of 5 unit cells (2 µm over a 400 nm lattice), we expect approximately $\frac{\pi }{6}5^{3}\approx 65$ unit cells within the confined volume and $\frac{\pi 5^{2}}{\sqrt{2}}\approx 55$ at the surface (the actual number of fractional unit cells according to the simulator is 52). Consequently, solving the unit cell becomes of limited use, and efficient computational approaches designed for three-dimensional photonic crystals [7] are not readily applicable in this scenario.

Structures with a complexity similar in magnitude to the wavelength are typically solved accurately and reliably using FDTD solvers. Representing the membrane as an FDTD structure is straightforward: the volume of the cubic membrane is converted into the solver. This conversion allows the entire spectral response to be proved with a single broadband pulse, eliminating the need for simulating the transmission for each monochromatic wavelength and subsequently combining the results.

However, the multilayer membrane consists of structures on the scale of one nanometer, resulting in enormous meshes. Even with a grid resolution of 2 nm, considerable deviations from the rigorous solution are observed. To assess the limitations and accuracy of the approach, we rigorously solved the reflection/transmission curve for a flattened, extended membrane of the original thickness [8] and compared it to the results obtained using FDTD. While the errors for a 1 nm mesh are acceptable, the stability of the method is not satisfactory.

Meshing a 2 µm mitochondrion results in an overwhelming number of approximately 2 billion mesh entries, pushing the limits of our computational capabilities. Nevertheless, we conducted simulations for the 2-micron case using both ray tracing and finite difference time domain (FDTD) methods. The discrepancies observed in the results highlight the limitations of both approaches, but they also qualitatively support our initial hypothesis. Thus, in this study, we focus on the ray tracing approach since it enables proper modeling of the enclosed mitochondrion without disregarding its boundary.

Considering the 12-layer membrane and the homogeneous interstitial compartments as fundamental building blocks, we proposed a straightforward solution by replacing the membrane structure with an equivalent homogeneous material. This substitution conveniently scales the cubic membrane of the mitochondrion to a size comparable to well-studied photonic crystals. However, rotating the membrane stack at various aspect angles reveals the non-existence of such a material. Similarly, replacing the membrane with the average refractive index yields no significant photonic properties. Thus, we can confidently assert that the cubic membrane present in the lens mitochondrion cannot be considered a proper photonic crystal. Instead, a reasonably accurate approximation can be achieved by replacing the membrane with a thin sandwich material characterized by steeply changing refractive indices (RIs). This adjustment brings the model closer to the scenario elegantly solved by Momeni et al. [7]. Moreover, it simplifies the mesh volume significantly and reduces its size.

Overall, these challenges and limitations arise when utilizing various simulation approaches to investigate the unique characteristics of the lens mitochondrion, including its confined size and complex structure. The computational constraints associated with meshing a 2 µm mitochondrion emphasize the need for careful consideration. However, by employing ray tracing, we were able to effectively model the enclosed mitochondrion while accounting for its boundary. To address these challenges, we explored alternative approaches and introduced a thin sandwich material with steeply changing refractive indices (RIs) as a promising approximation. This adjustment simplifies the mesh volume and demonstrates qualitative agreement with our initial hypotheses. We emphasize the importance of further refinement and exploration to achieve more accurate results in the study of lens mitochondria.

### 2.2. Closed Bilayer Approximation

For a well-defined multi-layer system with known materials and geometry, the reflectance dependents on the wavelength and angle (θ) of incidence.

To approximate the multi-layer, we minimized the reflectance difference between it and the closed bilayer. This involves optimizing the four parameters of the bilayer, denoted as: $X=\{n_{a},n_{b},d_{a},d_{b}\}$, which represent the outer reflective index, the core reflective index, and their respective thicknesses.
$$\underset{X=\{n_{a},n_{b},d_{a},d_{b}\}}{\min}\sum_{i,j}\left|R_{⊥}\left(\lambda_{i},\theta_{j}\right)-{\hat{R}}_{⊥}\left(\lambda_{i},\theta_{j}\right)\right|+\left|R_{∥}\left(\lambda_{i},\theta_{j}\right)-{\hat{R}}_{∥}\left(\lambda_{i},\theta_{j}\right)\right|$$
where $R_{⊥}=\left\|r_{⊥}\right\|$, and $R_{∥}=\left\|r_{∥}\right\|$ are defined by the reflection coefficients, depending on the direction of polarization.

The wavelength *λ* is sampled within visible light from 400 to 800 nm in 5 nm steps. The incident angle θ is sampled from 0 to 90 degree stepped in single-degree steps. The optimization algorithm “interior-point” was used to establish the parameters of the closest bilayer in Matlab.

The obtained results presented in the Table 1 demonstrate a satisfactory level of accuracy while also revealing some unexpected findings. Firstly, it can be observed that the average RI and overall thickness of the structure are maintained closely, indicating a successful approximation. Furthermore, the introduced sandwich structure exhibits improved performance in handling total internal reflection compared to the homogeneous material approach (Table 1).

Remarkably, it was discovered that the core material within the sandwich structure is identical to the interstitial material, effectively resulting in two photonic structures nested within each other. Although some maximum errors are experienced at high angles, the deviations remain within the low percentage range. The presence of thick sandwich layers allows for a coarser meshing approach compared to the original structure, offering computational advantages. Additionally, since the mitochondrion contains fewer than five repeats of the unit cell, errors at high angles do not accumulate superlinearly. To balance between accurately modeling the rigorous solution presented in Table 1 and to ensure computational feasibility within the memory constraints of our largest compute node, we employed a compromise meshing strategy. The curvature of the membrane necessitated smaller mesh cells of approximately 5 nm between material transitions, while no such limitation exists within homogeneous materials. As a result, the mesh size was reduced by about 2.5 orders of magnitude, and the entire simulation could be completed within a week of compute time for all wavelengths.

### 2.3. Alternative Approach

An alternative approach can be devised by neglecting diffraction effects and focusing solely on energy transport and polarization. This approach reduces the problem to a local ray-tracing method which sacrifices the accuracy of light distribution geometry but ensures localized energy preservation and rigorous treatment of interference contributions, limited by the numerical accuracy of sampling and integration (Figure 2). To implement this method, we employed a Monte Carlo process in which a finite number of rays, sharing the same frequency, polarization, phase, and direction (parallel to the *x*-axis), were emitted from a 2D grid in the y-z plane. The grid was arranged with offsets to cover a circular region centered around the y-z origin, matching the radius (R) of the mitochondrion. This arrangement guaranteed that all fired rays would intersect with the virtual mitochondrion.

The splitting condition for the Monte Carlo process was defined as the transition between different membrane compartments or the entry/exit points of the mitochondrion. The compartments were separated by a sandwich structure composed of lipid bilayers, with cytosolic layers of comparable thickness (each layer approximately 5–8 nm wide). In the tree shrew’s mitochondrion, there are 12 lipid bilayers separating the compartments. Although the sandwich wall between the compartments is not thin compared to the compartment volume (approximately 120+ nm versus a lattice of 400–500 nm), we opted to model the membrane transition “sandwich” as a locally flat stratified medium with 25 layers, including the contacting compartments. This simplification was possible due to the significantly smaller size of the individual layers compared to the wavelength.

This simplification greatly reduces the complexity of the problem by leveraging well-established investigations on transitions through stratified media [9]. We adopted Hecht’s simplification [8] to solve the transition at any angle, extending it to cover arbitrary polarization cases.

To achieve this, we divided our complex light packet into two components: a forward component with polarization in the plane of incidence onto the membrane transition and a lateral component with polarization orthogonal to that plane. Each fraction was treated independently, and their results were merged after the scattering process.

The Hecht solution considers each stratified layer as an element that alters the phase, direction of propagation, and fractions of propagated and reflected components. It describes this transition using a 2 × 2 matrix, $\left(\begin{array}{c} E_{I}\\ H_{I} \end{array}\right)=M\left(\begin{array}{c} E_{II}\\ H_{II} \end{array}\right)$ where index I marks the fields entering medium 1 from medium 0 and index II the fields entering medium 2 from medium 1. The direction of propagation of light enters the medium *a* (with thickness $d_{a}$ and a refractive index of $n_{a}$) at an angle θ and hence has the apparent thickness $h_{a}≔n_{a}d_{a}cos\left(\theta \right)$ in the direction of propagation.
$$M=\left[\begin{array}{cc} cos(k_{0}h_{a}) & i\;sin(k_{0}h_{a})/\gamma_{1}\\ i\;\gamma_{1}sin(k_{0}h_{a}) & cos(k_{0}h_{a}) \end{array}\right]$$

Here, $\gamma_{1}≔\sqrt{\frac{\epsilon_{0}}{\mu_{0}}}n_{a}cos\left(\theta \right)$ if the electrical field is parallel to the plane of incidence (polarization *s* state). When the electrical field is perpendicular to the plane of incidence (polarization *p* state), $\gamma_{1}≔\sqrt{\frac{\epsilon_{0}}{\mu_{0}}}\frac{n_{a}}{cos\left(\theta \right)}$.

Hecht’s solution demands the continuity of the tangential fields across an interface and strictly linear dielectric properties $H=\sqrt{\frac{\epsilon_{0}}{\mu_{0}}}nE$.

For a polarized electrical field in ***s*** state, the following equations will hold as a boundary condition,
$$\left[\begin{array}{c} \left(E_{i1}-E_{r1}\right)cos\left(\theta_{i1}\right)\\ \sqrt{\frac{\epsilon_{0}}{\mu_{0}}}n_{0}\left(E_{i1}+E_{r1}\right) \end{array}\right]=M\left[\begin{array}{c} E_{t2}cos\left(\theta_{t2}\right)\\ \sqrt{\frac{\epsilon_{0}}{\mu_{0}}}n_{s}E_{t2} \end{array}\right]$$

While in the *p* state,
$$\left[\begin{array}{c} \left(E_{i1}+E_{r1}\right)\\ \sqrt{\frac{\epsilon_{0}}{\mu_{0}}}n_{0}cos\left(\theta_{i1}\right)\left(E_{i1}-E_{r1}\right) \end{array}\right]=M\left[\begin{array}{c} E_{t2}\\ \sqrt{\frac{\epsilon_{0}}{\mu_{0}}}n_{s}cos\left(\theta_{t2}\right)E_{t2} \end{array}\right]$$
with $E_{i1}\;and\;E_{r1}$ representing the amplitudes incoming from the medium 0 to medium 1 and reflected at the medium 1, respectively, $E_{t2}$ representing the transmission into the next medium 2, $\;and\;\theta_{i1}\;and\;\theta_{t2}$ representing the angles of incidence into medium 1 and within medium 1 into the next medium, respectively. Finally, the refractive indexes of the medium before entering medium 1 are $n_{0}$ and the subsequent one, $n_{s}$.

The propagation through many layers is again given by the product of all matrices, and the transmission and reflection coefficients are defined by:$$r=\frac{E_{r1}}{E_{i1}}\;\mathrm{and}\;t=\frac{E_{t2}}{E_{i1}}$$

For a polarization *s* state, introducing the auxiliary ratios $\gamma_{o}≔\sqrt{\frac{\epsilon_{0}}{\mu_{0}}}n_{0}cos\left(\theta_{i1}\right)$ and $\gamma_{s}≔\sqrt{\frac{\epsilon_{0}}{\mu_{0}}}n_{s}cos\left(\theta_{t2}\right)$, the coefficients will be:$$r_{⊥}=\frac{\gamma_{0}m_{11}{+\gamma }_{0}\gamma_{s}m_{12}-m_{21}-\gamma_{s}m_{22}}{\gamma_{0}m_{11}{+\gamma }_{0}\gamma_{s}m_{12}+m_{21}+\gamma_{s}m_{22}}$$
and
$$t_{⊥}=\frac{{2\gamma }_{0}}{\gamma_{0}m_{11}{+\gamma }_{0}\gamma_{s}m_{12}+m_{21}+\gamma_{s}m_{22}}$$

While in the *p* state, setting $\gamma_{o}≔\sqrt{\frac{\epsilon_{0}}{\mu_{0}}}\frac{n_{0}}{cos\left(\theta_{i1}\right)}$ and $\gamma_{s}≔\sqrt{\frac{\epsilon_{0}}{\mu_{0}}}\frac{n_{s}}{cos\left(\theta_{t2}\right)}$, the coefficients will become:$$r_{∥}=\frac{{-\gamma }_{0}m_{11}{-\gamma }_{0}\gamma_{s}m_{12}+m_{21}+\gamma_{s}m_{22}}{\gamma_{0}m_{11}{+\gamma }_{0}\gamma_{s}m_{12}+m_{21}+\gamma_{s}m_{22}}$$
and
$$t_{∥}=\frac{{2\gamma }_{0}cos\left(\theta_{i1}\right)}{\left(\gamma_{0}m_{11}{+\gamma }_{0}\gamma_{s}m_{12}+m_{21}+\gamma_{s}m_{22}\right)cos\left(\theta_{t2}\right)}$$

This addition accounts for the parallel polarized case, which is the only extension we made over Hecht’s approach. The general polarization is split into these two components at each scatter process, and after reflection and transmission, both components are merged again. Therefore, the scatter process incorporates the possibility of a polarization change.

The ratio of transmitted and reflected power is deterministic and depends on factors such as the wavelength, incident angle, polarization, and refractive index change. However, the diffractive process is simulated using a stochastic component that accounts for the variability in the propagation direction. Instead of directly simulating the propagating wave, this approach involves substituting it with one or several rays. These rays’ directions and relative powers are modeled based on the probability of propagation. This simulation technique resembles how incident light decomposes from a continuous wave into individual photon interactions.

Thus, in the stochastic process, the transition is assumed to occur precisely between the two compartments and is considered infinitely thin. However, diffraction, polarization change, reflection, and phase alteration are still calculated based on the original structure’s width. Thus, the transition acts as a black box that transmits, bends, reflects light, and modifies its polarization depending on the wavelength, polarization, and angle of incidence. However, the stochastic splitting is computed only once per transition.

We used the same physics as described in [6], but we no longer relied on the simplifications concerning the *z*-axis and polarization. Additionally, at this stage, we did not need to consider shape inaccuracies or tolerances. Therefore, our pure interference model was limited to a single transition, and we employed scattering between transitions to transport the electric field from one event to the next. Total internal reflection can occur at either the interface of the first layer or at the exit from the last layer, and we verified its occurrence. However, we simplified all cases by using the center approximation, as depicted in Figure 3. When total internal reflection occurs, no interference transition is calculated, and it is always reduced to a perfect reflection.

We explored several mechanisms for implementing the scattering process, referred to as “photon splitting”:

(i) Only the transmitted or reflected portion survives, with a probability based on the relative energy of the two fractions. This ensures algorithm termination and requires a large number of initial rays for numerical accuracy.

(ii) Both portions are pursued, starting with the weaker portion. If a portion’s energy falls below a fraction ϵ (chosen as one billionth of the initial energy), it is discarded. This approach uses minimal memory as the weaker portion loses energy exponentially, and it provides stable results with a fraction of the original rays. We found convergence with 4000 or more original rays, and the fraction of lost “photons” can be controlled by adjusting ϵ to keep the total lost energy below 1%.

(iii) Splitting the scattered portion into multiple packets to account for diffraction. However, this approach did not yield smoother outcomes or require fewer rays compared to case (ii) in the small set of transitions tested. Consequently, we abandoned this approach in favor of faster computation.

### 2.4. Simulation Models

In order to simulate diffraction, we employed a distribution function resembling a confined parallel wave, with lateral confinement determined by the average separation of our sample rays. We disregarded the impact of the finite width and high curvature of the membrane transition in this general case. Implementing approach (ii) for this scenario, we observed that the distribution function of the propagated “rays” did not significantly affect the quantitative results; it primarily affected the smoothness of the output when densely sampled (using one ray per 2 to 10 nm). This lack of notable contribution can be attributed to the highly convoluted surfaces and the limited scale over which diffraction occurs, which governs the scattering process. Therefore, for most of our results, we neglected small-scale diffraction and relied on the pure geometric propagation of the electric field packets for the narrow confinement. In order to address the merging and interference phenomena of the transmitted light, we employed two distinct approaches to approximate the near-field characteristics illustrated in Figure 4 and Figure 5.

The computational effort required for the model depicted in Figure 5 does not significantly increase when compared to model depicted in Figure 4 for two reasons: (a) the scattering events account for the majority of the computing time, and (b) the number of surface tiles can be kept low, considering the small dimensions of the mitochondrion. If 1/4λ patches are used some 100 nm in diameter on the *x* axis, using angular $\left(\theta ,\vartheta \right)$ tiling of the surface, ${\left(\pi \frac{1\mu m}{100nm}\right)}^{2}\sim 1000$ tiles are required, leading to the same amount of integration steps per pixel per image plane.

As the exact true (in vivo) refractive indices of the compartments and lipid bilayers, which constitute the membrane separation in addition to the cytosolic layers, could not be determined, we relied on the published refractive indices of the bulk materials present in these volumes [10]. The specific refractive indices of the two compartments of the lens mitochondrion are also unknown. However, considering that the focal length (for which we used the distance to the photoreceptor due to lack of experimental data) of the lens mitochondrion is shorter than it would be for a cytosolic material with the given fraction of lipid bilayers, it is likely that at least one of the compartments has a higher refractive index. Based on models derived from TEM projections, we estimate that each major compartment occupies approximately 29% of the unit lattice, while cytosolic layers and lipid bilayers each occupy around 21%. If one compartment is cytosolic with a refractive index of approximately 1.33, the second compartment could have a higher refractive index, resulting in more intriguing outcomes. The average refractive index of the entire mitochondrion was chosen to enable diffraction and focus light on the photoreceptor, although it remains uncertain if this is indeed the case. We tested refractive indices ranging from 1.38 to 1.48 for the compartment with the higher refractive index, all of which fall within the plausible range for the average refractive index, although the specific values for each compartment remain unknown. The wavelength range of 350–900 nm was tested in our simulations. Firstly, this range covers the visible spectrum, which is relevant for the study of the optical phenomena of the lens mitochondria, and secondly, the selected range allows for a comprehensive exploration of the behavior of light in the system, capturing both shorter and longer wavelengths that may exhibit distinct characteristics and effects.

## 3. Results and Discussion

The close resemblance between the internal membranes of the lens mitochondria in Tupaia species and certain photonic crystals with photonic bandgap (PBG) properties is unmistakable [6,11]. Photonic crystals are periodic composite nanostructures that manipulate the propagation of electromagnetic waves, much like how a periodic potential in a semiconductor crystal governs the movement of electrons by establishing permitted and forbidden electronic energy bands [12]. The simplest form of a photonic crystal is a one-dimensional periodic structure, such as a multi-layer film, of photons can be achieved through the use of three-dimensional periodic crystals with PBG, which prohibit the transmission of light waves in any direction and for any polarization within a specific frequency range. With both lens mitochondria and photonic crystals with PBG having similar geometries (gyroid type), it is tempting to conjecture that the cubic membrane architecture in the lens mitochondria of Tupaia may act as a photonic crystal with a lattice size similar to ultraviolet wavelengths and that its PBG properties may block or direct UV light from reaching the outer segments of photoreceptors in the rod-deficient retina.

In this study, we only considered the light that actually traversed the mitochondrion, i.e., packets which fulfilled $\left\|e_{0}+s\cdot e_{p}\right\|=R_{0}$, with $e_{p}$ being the direction of initial propagation. All results were generated under this restrictive assumption.

### 3.1. Focus and Dispersion

In the near field, the results obtained from the approach depicted in Figure 5 show minimal differences compared to the approach shown in Figure 4. Neither of the models exhibited any discernible caustics that could provide a more precise resolution. The interference mechanism employed in both models disrupts the geometric focus and introduces a wavelength-dependent shift of the focus, resembling dispersion effects, despite the absence of explicit dispersion models at this stage. Notably, when considering RI value of 1.48 for the second compartment, this wavelength-dependent shift becomes highly noticeable.

The position of the geometric focus is wavelength-dependent, resulting in a variation in its location (Figure 6). Short wavelengths tend to be focused closer to the lens mitochondrion, while long wavelengths are focused further away. All focal points reside along the long axis of the folded membrane in the outer segment of the retina. However, it should be noted that for wavelengths of approximately 800 nm (twice the lattice constant), the geometric focus is either lost or substantially weakened.

### 3.2. Waveguide Properties

As anticipated, when the two compartments possess different refractive indices, the channels within the compartment with the higher refractive index act as a mild waveguide, preferentially guiding light within that particular compartment, as suggested by Kucki et al. [13]. The projection plot (Figure 7) illustrates that the angle for capturing light into these channels is surprisingly large, and the efficiency of waveguide transport increases with the wavelength of the light. The capturing angle is challenging to explain solely as a result of a simple total internal reflection waveguide as the size of the membrane separation is too large. When the mitochondrion is oriented in such a way that the channels no longer capture light, this phenomenon diminishes accordingly (Figure 8).

The waveguide exhibits a transmission loss of a few dB per meter. However, due to the small size of the structure (approximately 5 μm), we can safely assume that attenuation and absorption are negligible. Consequently, the filtering effect of the waveguide primarily arises from scattering and reflection.

The entire system demonstrates robustness against rotation and changes in direction. Although the shape of the focal spot undergoes dramatic changes based on the transmission direction, the power of the transmission remains consistent. Consequently, the structure can be described as a quasi-bandgap or an imperfect photonic crystal, as depicted in Figure 9 and Figure 10.

TEM images often depict an apical lens mitochondrion positioned on top of three to six closely packed mitochondria with similar membrane organizations, forming a solid mass that fills the conical upper region of the inner segment of the photoreceptor in Tupaia belangeri [4]. The TEM images also show multiple points of contact between the densely packed mitochondria and a consistent cubic membrane orientation across adjacent mitochondria [4]. These observations from the TEM images suggest that mitochondria with a cubic membrane arrangement may have waveguide properties that direct light from the inner mitochondrial layer toward the apical mitochondrion, where it could be further focused onto the narrow outer segment of the photoreceptors. Our data support this idea, demonstrating that the channel network within the compartment with a higher refractive index forms a mild waveguide that preferentially directs light within that compartment.

### 3.3. Loss of Focus

For wavelengths close to 800 nm (twice the lattice constant), the geometric focus is either lost or significantly weakened. However, for wavelengths above 650 nm, a second elongated region emerges where the light is relatively focused (referred to as the “2nd focus”). In this region, the power flux surpasses that of the lost focal region. This phenomenon is specifically observed for wavelengths around twice the lattice constant (Figure 11). The biological significance of this second focus region remains mysterious.

This multi-focal property of lens mitochondria may improve the color perception of tree shrews, which are diurnal animals with high metabolic rates and voracious appetites. They consume up to 80–90% of their own body weight in food daily. Without trichromacy, these tree shrews would have the same difficulty as mammalian dichromats in detecting ripe fruit against the background of forest foliage [14,15]. This is because the spectral reflectance curves of yellow or orange fruit differ from those of the leaves, primarily at wavelengths above 500 nm. As a result, a dichromatic consumer cannot detect such fruit by color because their short-wave cones, which absorb negligibly above 500 nm, are responsible for color vision.

### 3.4. Interference Filter

With one compartment showing an elevated RI, the lens mitochondrion becomes an imperfect though quite effective UV-shielding interference filter.

The peak ridge depicted in Figure 12 is distinct from the previously discussed geometric focus and consistently appears behind that region. Furthermore, the position of this peak is influenced by the rotation of the mitochondrion and slight variations in the refractive index (RI), whereas the overall power is less affected. Both the peak position and power are sensitive to RI alterations in the membrane transitions.

Our simulations of the transport of light through the multi-layered cubic membrane arrangement suggest that the photonic crystal properties may not be necessary for the lens mitochondria to act as a wide-angle, broad-band UV filter in front of the outer segment of photoreceptors. Our simulations also indicate that a simplified assembly of such membranes has the power to disperse UV light and reduce its transmission by up to 50%. No definitive focus points were identified for wavelengths less than 400 nm or more than 800 nm. The interference filters also have sufficiently steep transitions to allow visible light to reach the eye (Figure 12). The lens mitochondria may not function as a perfect quasi-photonic crystal due to the small changes in the refractive index and the narrow lipid-bilayer-based membrane (approximately 5 nm), but it serves as an effective bandpass that suppresses the majority of short wavelengths while efficiently transporting longer wavelengths through the multi-layered cubic membrane arrangement.

Excessive exposure to UV can have detrimental effects on neurons [16]. UV radiation has been shown to induce oxidative stress and generate reactive oxygen species within cells, leading to damage to various cellular components, including DNA, proteins, and lipids [17]. In addition to damaging intracellular molecules, UV can also affect other intracellular organelles, including the mitochondria [18]. It has been reported to cause alterations in endoplasmic reticulum (ER) function and calcium homeostasis, leading to ER stress and unfolded protein response activation. UV-induced ER stress can further contribute to neuronal dysfunction and cell death [19]. Furthermore, UV can impact lysosomal function and autophagy, processes crucial for cellular waste clearance and protein quality control.

Organisms, including animals and humans, have developed various protective mechanisms to shield themselves from the harmful effects of UV radiation [20]. The skin serves as a barrier and produces melanin, a pigment that absorbs UV radiation and reduces its penetration into deeper layers of the skin [21]. Animal cells have also evolved specialized mechanisms to counteract UV-induced damage. They possess antioxidant defense systems, e.g., enzymes such as superoxide dismutase and catalase, which help neutralize the reactive oxygen species generated by UV exposure [20]. Additionally, DNA repair mechanisms, including nucleotide excision repair, operate in animal cells to correct UV-induced DNA damage and maintain the integrity of the genome [22]. These adaptive strategies and cellular mechanisms work together to minimize the detrimental effects of UV radiation on animal cells. In the case of Tupaia species, our simulation data provide insights that could explain experimental findings demonstrating the retina’s ability to block the transmission of near-blue and UV light to the outer segment [23]. These findings suggest that the unique lens mitochondria observed in Tupaia may play a crucial role in this protective mechanism.

## 4. Conclusions

This study aimed to investigate the transmission of light through the lens mitochondria in the optical systems of small mammals, specifically the tree shrew Tupaia. Our simulation of light transport through these multi-layered lens mitochondria with cubic membrane morphologies revealed several potential functions. Firstly, the lens mitochondria may act as a multi-focal lens capable of focusing light at different points. Secondly, they may function as angle-independent interference filters, effectively blocking UV light. Finally, the mitochondria demonstrate characteristics of waveguide photonic crystals. These findings hold true regardless of the integration model used in the simulator, although any faults in the scattering mechanism would impact both models equally. It is worth noting that the computational effort required for the scattering part of the simulation is substantial. Despite employing a fully parallel version implemented in CUDA [24], the runtime for a single wavelength exceeded 1500 **s**. While the numerical outcome is acceptable, a lower number of rays may exhibit visible granularity. Experimental validation poses a challenge as constructing a multispectral transmission microscopy setup is more complex than the simulator itself. The illumination with a low-numerical-aperture plane wave may introduce the diffraction of non-transmitted light around the mitochondrion, potentially contaminating the readout. Our best hope lies in modifying the simulator setup to account for the high numerical aperture of a point scanner or the confined output of a fiber illuminator. However, to date, we have not obtained experimental confirmation of our findings.

The authors acknowledge the relevance of the temperature dependence of the coating; however, due to current limitations, access to data on this dependency is unavailable. The observational data used in this study are based on fixed electron microscopy samples, which eliminates any direct relationship to temperature. Consequently, the simulation can only provide partial insights into this aspect. While the expansion of structures with temperature is well understood and can be easily simulated, its impact is relatively minor. The changes in the refractive indexes of the materials are only known for laboratory temperatures within the range of approximately 20–25 degrees Celsius. A future experiment utilizing actual mitochondria may offer some insights into temperature dependency. However, it is important to note that mitochondria exist within mammals, where they experience a tightly regulated temperature of approximately 37 degrees Celsius. Since the publication does not make claims regarding the ex vivo function of mitochondria, addressing the temperature dependency in this specific context falls beyond the scope of this study.

This study has potential applications in optics and photonics, biomedical imaging, and ophthalmology and vision research, as well as biomimicry and bioinspired design. The findings can contribute to the development of novel optical devices, improve imaging techniques, enhance eye protection against UV damage, and inspire innovative engineering solutions. The unique properties of lens mitochondria offer opportunities for advancements in these fields and may lead to the creation of improved optical materials and technologies.

## Figures and Tables

**Figure 1 membranes-13-00610-f001:**
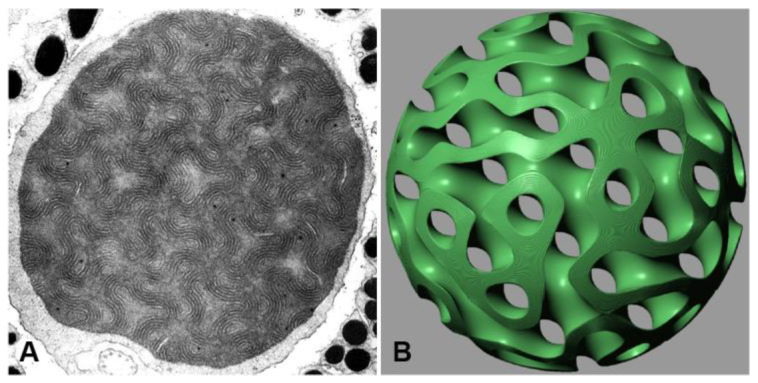
(**A**) TEM micrograph of lens mitochondrion observed in the retinal cones of tree shrew species, which is revealed to be six pairs (12 layers) of gyroid-based parallel level surfaces [5]. (**B**) Mathematical 3D model that can be used to describe the multi-layer gyroid cubic membrane arrangement identified in lens mitochondria. The original TEM micrograph in (**A**) is adopted with permission from [4].

**Figure 2 membranes-13-00610-f002:**
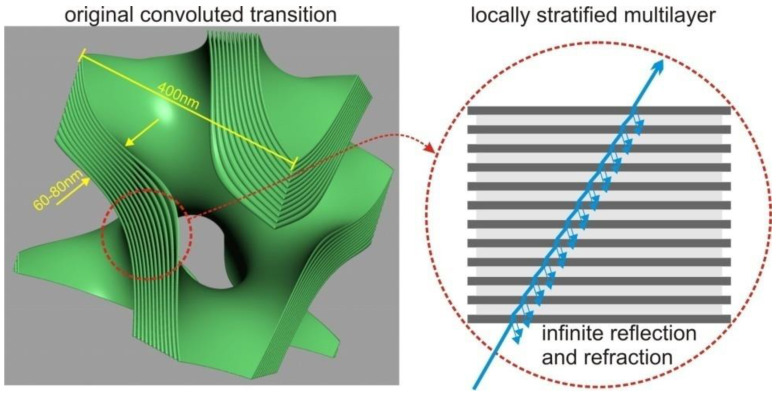
Depiction of the local flatness simplification, which allowed us to use a stratified media interference structure for the scattering process.

**Figure 3 membranes-13-00610-f003:**
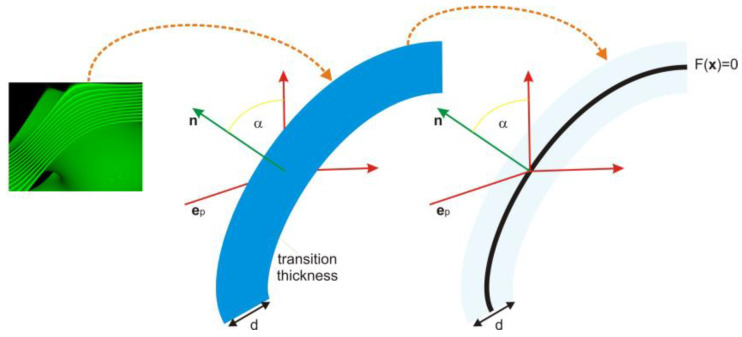
Illustration of the process of simplification wherein the complex multilayer membrane transition is condensed into a single element known as the stratified ”sandwich”. Subsequently, collision detection and scatter event calculations are focused solely on the core of this sandwich structure.

**Figure 4 membranes-13-00610-f004:**
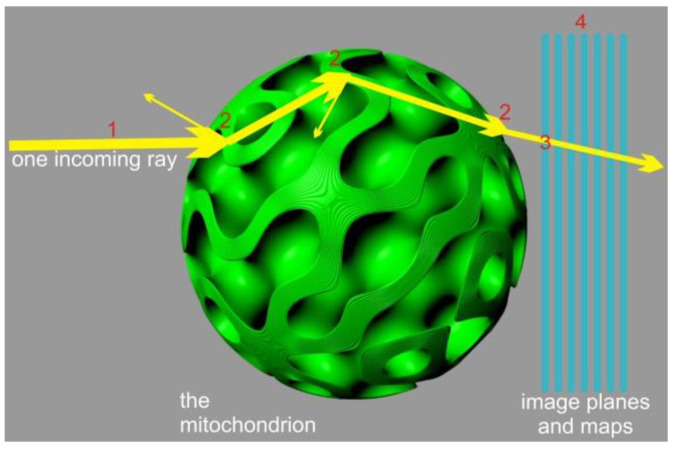
First-order diffraction model. The model shows probing a significant number of rays (1) and accounting for scattering events geometrically (2). The transmitted rays were accumulated in high-resolution accumulator maps, considering their phase, amplitude, polarization, and direction (3). These maps were then integrated with a diffraction-limited point spread function onto the image planes (4). The resulting model in the image planes represented a purely geometric approximation of the near field (here, a cubic membrane with only 6 lipid bilayers is shown for illustration).

**Figure 5 membranes-13-00610-f005:**
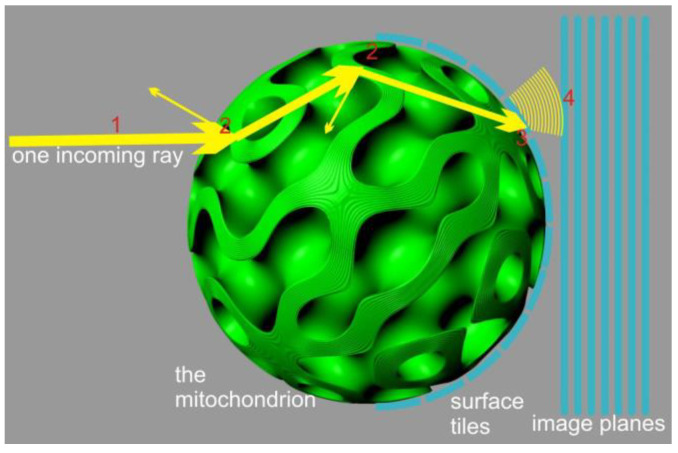
The unknown order diffraction model. A significant number of rays are probed (1), and the scattering events within the mitochondrion are tracked geometrically (2). Then, in an initial integration step, surface patches gather information regarding phase, amplitude, and vector of the ray hits (3). Finally, in a subsequent integration step, the surface patches emit their contributions to the image planes (4).

**Figure 6 membranes-13-00610-f006:**
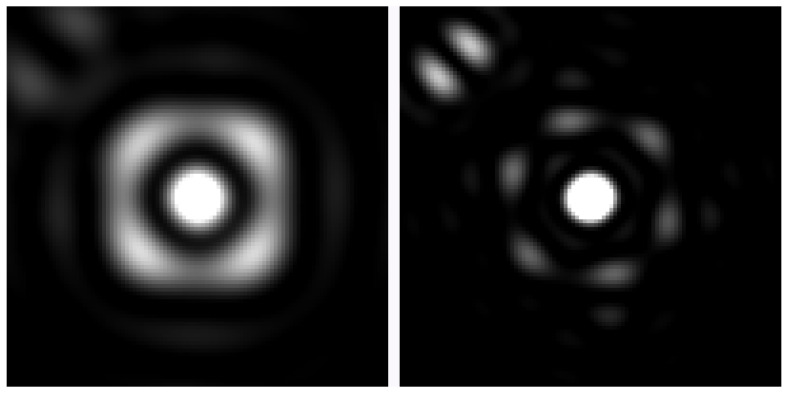
Quasi-dispersion phenomenon, showcasing the region where the best geometric focus occurs. In the case of 600 nm light, the area of optimal focus is located approximately 3.3 µm behind the mitochondrion (displayed on the left side of the figure). Conversely, for 350 nm light, the region of optimal focus is found approximately 2.6 µm behind the mitochondrion (depicted on the right side of the figure). The diameter of the panels in the figure is 2 µm.

**Figure 7 membranes-13-00610-f007:**
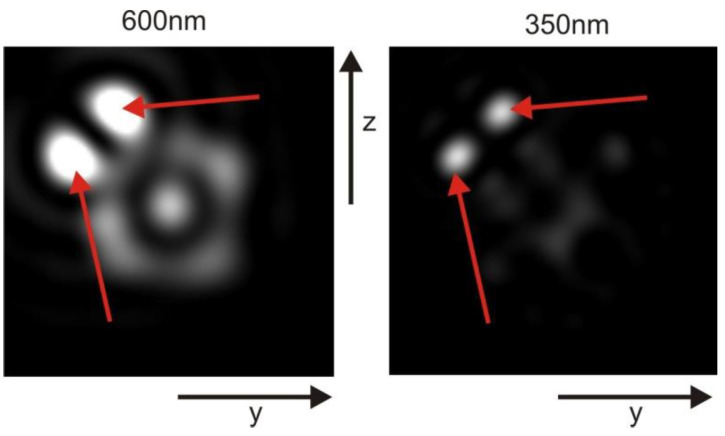
The figure illustrates the emergence of ‘holes’ that become visible under specific angles, forming waveguides. On the right side of the figure, the near-field light originating from these waveguides is depicted.

**Figure 8 membranes-13-00610-f008:**
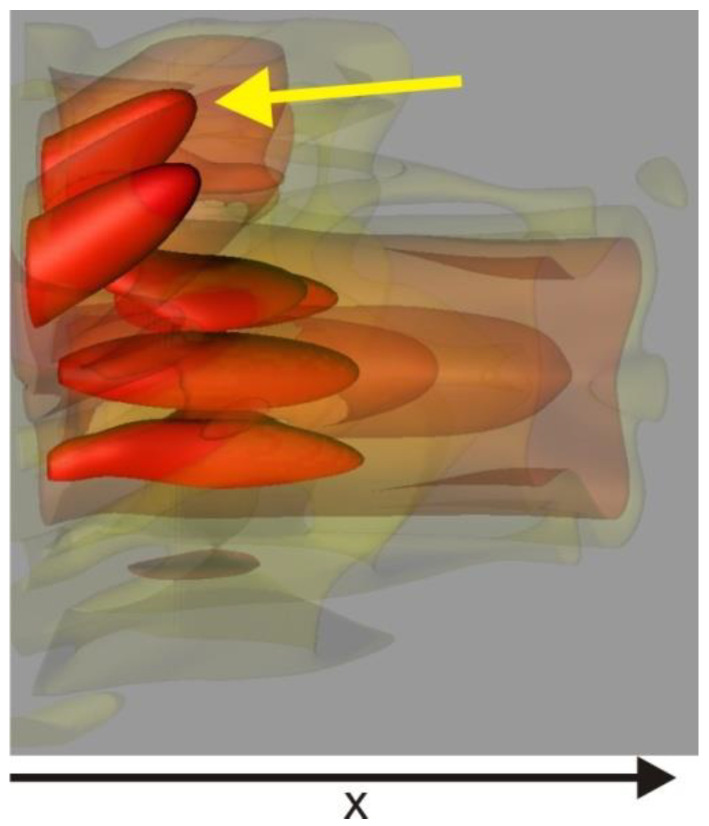
Iso-intensity 3D plot of the intensity pattern after passing through the mitochondrion. The yellow arrow highlights the peaks that arise from the waveguide property.

**Figure 9 membranes-13-00610-f009:**
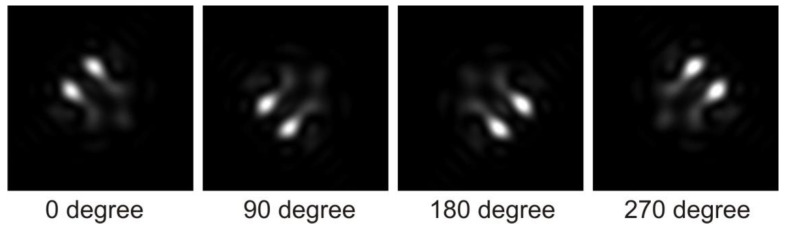
The figure illustrates the trivial symmetry observed when the structure is rotated in multiples of 90 degrees around any axis. The intensity and waveguide pattern remain unchanged except for orientation.

**Figure 10 membranes-13-00610-f010:**
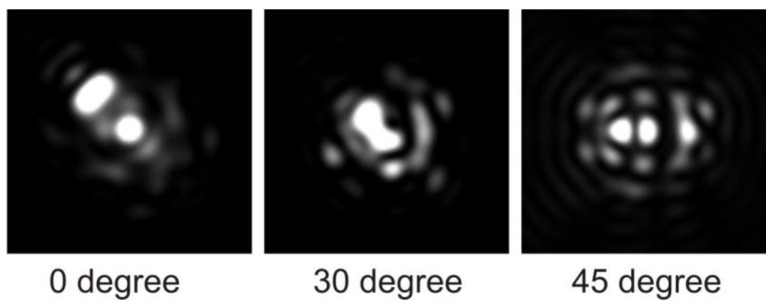
The figure demonstrates the significant changes in the pattern at the focus for different values of the rotation angle. Additionally, the intensity within the central beam exhibits fluctuations of approximately 41%, while the total transmitted energy only varies by around 9%, depending on the angle of incidence (for RI = 1.48).

**Figure 11 membranes-13-00610-f011:**
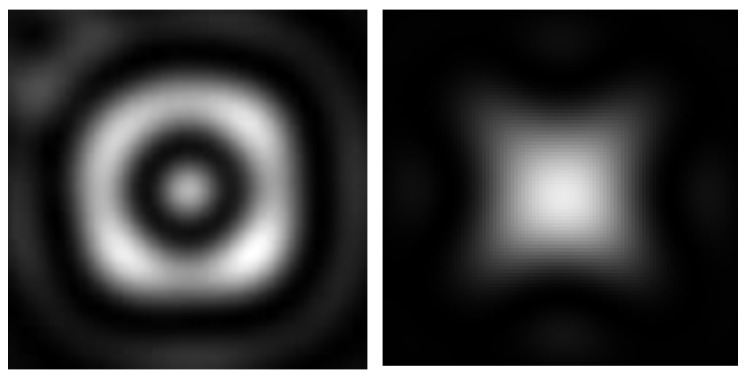
The figure illustrates the loss of geometric focus for 800 nm light. Even in the area of the lowest confusion and highest intensity, the peak intensity is no longer centered (the peak is located 3.5 µm from the lens mitochondrion). The second focus is plotted at 10 µm for comparison purposes.

**Figure 12 membranes-13-00610-f012:**
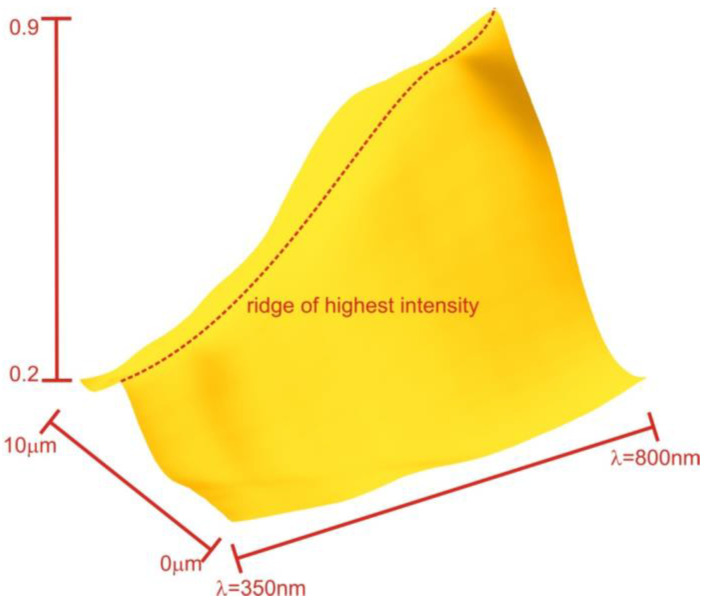
The figure shows the total power flux through a disk with the same diameter as the mitochondrion in the image planes behind the mitochondrion. The flux is presented as a ratio compared to the flux in the absence of mitochondria. The plot reveals that for most wavelengths, there is a specific plane on which constructive interference is most pronounced, resulting in the peak ridge indicated in the plot.

**Table 1 membranes-13-00610-t001:** Bilayer approximation. The RIs provided are referenced at the Fraunhofer D line, specifically at 587.56 nm. The wavelength dependence of the RIs or Abbe numbers of the materials, however, remain unknown.

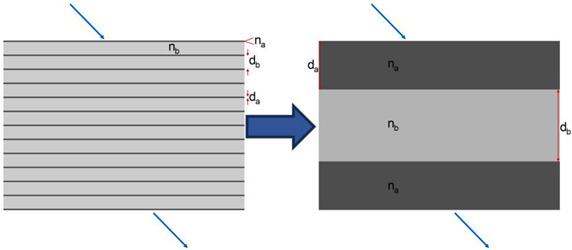			
	**12 Layers**	**Approximated Bilayer**	**Bilayer**
$n_{1}$	1.33		
$n_{2}$	1.33		
$n_{a}$	1.70		1.5536
$n_{b}$	1.38		1.33
$d_{a}$ (nm)	1.8		32.8
$d_{b}$ (nm)	10		48.6
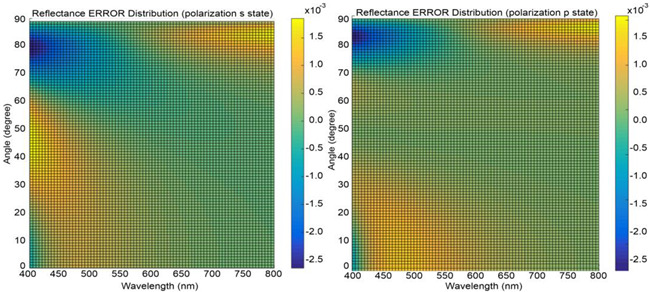			

## Data Availability

We would like to clarify that the data in our study primarily consists of mathematical equations product which are used in the analysis rather than empirical or experimental data. The equations employed in our study are provided within the main body of the article and can be referenced accordingly. These equations serve as the basis for our theoretical framework and enable the interpretation of the results presented.

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
