# Peer review of "Optical Properties and Interference Effects of the Lens Mitochondrion"

_membranes, 2023, doi:10.3390/membranes13060610_

Round 1

Reviewer 1 Report

Publication is recomended,

Author Response

We thank the reviewer for the positive opinion regarding our paper. 

We revised the manuscript and made the necessary amendments.

Reviewer 2 Report

Advanced Simulation of Light Transport in Cubic Membranes Using First Order Diffraction and Interference

Authors in this paper have devised a Monte Carlo method, where a finite but large number of rays with the same frequency, polarization, phase, and direction are fired from a 2D grid in the yz plane, to simulate the energy transport and polarization of a plane light wave through a lens mitochondrion of a Tree Shrew. They used Hecht's solution to solve the transition of the light through the stratified layers of the membrane, treating each layer as an element that changes the phase, direction of propagation, and fractions of propagated and reflected energy, and it describes this transition by a 2x2 matrix. This results of the simulation strongly suggest that the lens mitochondrion forms a quasi- bandgap or an imperfect photonic crystal. The interference mechanism destroys the geometric focus and introduces a wavelength-dependent shift of the focus, which looks comparable to dispersion.

I went through the paper very carefully and thoroughly and I have some comments:

1-     The paper contains interesting sciences especially how we can control the light. The impact of the paper on these results is going be good. Also, the quality of the research work presented in the paper is also good.

2-     In general, ideas are well explained and understandable but, some tenses, linkers and grammar structures must be checked.

3-The authors should give the thickness and number of layers of all layers that they calculated. Are these parameters obtained from an optimization process?

4. Authors should obtain the novelty of this manuscript compared to published results?

5. The authors should argue about the relevance of the temperature dependence of the coating.

6. The Introduction does not provide sufficient background. The introduction does not explain the major contributions and novelty of this work. The significance of the proposed solution has not been summed up.

7- The constructive discussions are missing. As mentioned earlier, authors must make a comparative analysis with other similar solutions and back up their claims on how the proposed solution can be considered as high performing compared to others.

8- If authors can consider the time variable in this study may be is better because the speed of light depend on the time.

9-How their results will be affected if they include energy loss in layers.

10- The novelty of this work should be stated explicitly in the text of the manuscript so that readers can get it easily.

11- Is TMM are appropriate in this kind of study?

12- Authors should check and confirm all equations with references.

13-  Authors should compare their results with the published data and different results.

14- There a lot of published papers in this field, authors should be explained the new in these results in sensors.

15- Is it possible if authors can  study before and after the Brewster angle for your materials interface because the significance of the incidence angle in your study with explaining the physical meaning.

16-Is there are difference between TE and TM modes in this study?

17- What are the kinds of materials which used in lens and its applications?

18- What is the reference wavelength?

19- Authors should explain one or two applications to their work.

20- All figures, symbols, and units should be improved.

21- It seems the title need revision by authors to become more informative.

22- Authors mentioned” For the stochastic process, the transition is assumed to be exactly between the two……………………” What do you mean by stochastic in your study?

23- Are every term and structure in the proposed design should be clearly and correctly presented not to mislead the reader.

24- If it is possible to compare between the computational and experimental results.

25-Finally, I recommend that the paper should be revised taking care of the above comments.

I wish to resend this paper after corrections and revise my comments

Author Response

1-     The paper contains interesting sciences especially how we can control the light. The impact of the paper on these results is going be good. Also, the quality of the research work presented in the paper is also good.

We thank the reviewer for the valuable feedback and positive comments regarding our paper.

2-     In general, ideas are well explained and understandable but, some tenses, linkers and grammar structures must be checked.

We thank the reviewer for the valuable feedback. We revised the manuscript and made the necessary amendments.

3-The authors should give the thickness and number of layers of all layers that they calculated. Are these parameters obtained from an optimization process?

Many thanks for the clarification. The number of layers is given by the species of the specimen under investigation. 6 pairs or 12 bilayers are common. Here we focus exclusively on the more complex and convoluted mitochondria that are known to participate in the optics of the eyes of some mammals. The thickness is taken from the electron microscopy micrographs, one of which is included with this paper.

  1. Authors should obtain the novelty of this manuscript compared to published results?

The incremental novelty over the referenced finite difference time domain methods is mentioned in the manuscript: our method can work with ways less compute time memory and complexity and a much coarse simulation mesh at the expense of computational accuracy. Over pure ray-tracing approaches, our methods do at least address diffraction and local interference phenomena. We point this out more clearly in the manuscript.

  1. The authors should argue about the relevance of the temperature dependence of the coating.

We have currently no access to this dependency. The observational data is from fixed EM samples and hence the relationship to temperature has disappeared and the simulation can only partially address this: the expansion of the structures with temperature is well understood and can be simulated trivially but is of small effect. The change of refractive indices of the materials is only known for the lab temperatures of some 20-25 Celsius.

A future experiment with actual mitochondria might partially address the temperature dependency. However, the mitochondria are in mammals and hence in a very narrowly regulated 37 Celsius environments. Since we do not make claims of their function ex-vivo, we feel this exceeds the scope of this publication. We have incorporated a new paragraph in the manuscript to provide further clarification on this point.

  1. The Introduction does not provide sufficient background. The introduction does not explain the major contributions and novelty of this work. The significance of the proposed solution has not been summed up.

We thank the reviewer for the feedback, and we made the necessary improvements to address the mentioned concerns.

7- The constructive discussions are missing. As mentioned earlier, authors must make a comparative analysis with other similar solutions and back up their claims on how the proposed solution can be considered as high performing compared to others.

While this is a very valid critique, this method was specifically developed to close a gap in the scope of the FDTD and ray tracing methods that both do not cover that domain. So we cannot address the issue for this particular application. We have mentioned the superiority of FDTD methods in the realm of photonic crystal work. We could – if that helps – outline the superiority of ray tracing methods in the realm of macroscopic applications.

8- If authors can consider the time variable in this study may be is better because the speed of light depend on the time.

Yes, indeed, in both FDTD and in ray tracing, the time progresses. In FDTD one usually chooses a fixed time-step whereas in our method, light travels until it “hits” the next material interface. The time needed is calculated and used to determine how much the phase of the light wave has changed. When all light waves are combined again, this allows us to calculate the total interference pattern and energy.

9-How their results will be affected if they include energy loss in layers.

The solution becomes dramatically easier! Currently, we have to take into account factions of rays that carry one billionth of the original energy in order to computer the interference credibly. If the sample had attenuation, then the rays with less than a thousand or so of power will vanish, making the simulation much, much faster.

The simulator has attenuation code built in for this scenario. The problem here is that the lens mitochondrion is almost pigment free and very clear, much like a human lens or cornea.

10- The novelty of this work should be stated explicitly in the text of the manuscript so that readers can get it easily.

We thank the reviewer for the feedback, and we made the necessary improvements to address the mentioned concerns.

11- Is TMM are appropriate in this kind of study?

Not entirely, because it does ignore diffraction contributions. Hence, we estimated the errors from the transfer matrix simplification on simple test objects which justifies its use. The arguments why it still can be used to a limited extend is that all diffraction processes happen in a space that is just mildly larger than the wavelength and that our optical target – the light sensitive rod – is just beyond the mitochondrion so that far field focussing is fairly irrelevant. We also tested a diffraction model for the transport from the mitochondrion to the retina (figure 5) and use that.

12- Authors should check and confirm all equations with references.

The handful equations we sourced are referenced. Most of the equations are derived by us and hence a reference could be confusing.

13-  Authors should compare their results with the published data and different results.

As outlined in point 7, the exact case is not yet addressed by competitive methods. We outlined well studied, primitive structures for which our method and FDTD produce very similar results, pointing to a basic equivalence – in the sense of underlying physics – of broadly accepted approaches and our simplification. There is likely a pathway to a future experiment that could settle remaining doubts but as of today, no experimental data does exist.

14- There a lot of published papers in this field, authors should be explained the new in these results in sensors.

Indeed, there is ample research in the field of photonic crystals. But as outlined by us, biological membranes are not photonic crystals proper. They are neither large in scale compared to the wavelength nor do they have regular outlines. Solving the wave propagation through them is hence a challenge, as a brute scale approach is very computationally expensive. We do reference to the cases that have been solved by existing methods. We make no claims with regards to the sensors and hence cannot comment on them other than it would likely make for a fascinating research topic.

15- Is it possible if authors can  study before and after the Brewster angle for your materials interface because the significance of the incidence angle in your study with explaining the physical meaning.

Cell membranes in general and mitochondria in particular are highly curved with respect to the wavelength of visible light (they are usually 5-20 wavelengths in diameter). Brewster angles do occur on a small scale on numerous occasions. The simulator handles them with 100% forward propagation and 0% back reflection. They are hence just a special case of forward transmission. Much more challenging is total internal refection that traps the captured light inside a membrane.

16-Is there are difference between TE and TM modes in this study?

No. We use the micrographs exclusively to determine the structure sizes and the membrane geometry. This particular paper does not interpret the EM image further.

17- What are the kinds of materials which used in lens and its applications?

Lens mitochondria are predominantly composed of phospholipid biomolecules, similar to other biological membranes. To date, the specific function of lens mitochondria remains unclear, and as a result, there are no specific applications attributed to them.

18- What is the reference wavelength?

The visible range of light. We tested 350-900 nm in our simulations. We made the necessary changes in the manuscript to address the mentioned point.

19- Authors should explain one or two applications to their work.

The application is that we now have the capability to simulate the light transport and focusing properties of large and convoluted mitochondria. Information that is currently inaccessible to experimental investigation.

The explanation delivered here is that these mitochondria protect the retinae from noxious ultraviolet radiation. We made the necessary improvements in the manuscript to address the mentioned point.

20- All figures, symbols, and units should be improved.

We thank the reviewer for the valuable feedback. We revised the figures, symbols, and units and made the necessary amendments.

21- It seems the title need revision by authors to become more informative.

We thank the reviewer for the feedback. We revised the manuscript title.

22- Authors mentioned” For the stochastic process, the transition is assumed to be exactly between the two……………………” What do you mean by stochastic in your study?

The ratio of the transmitted and reflected power is deterministic and depend on the wavelength, incident angle, polarization, and the refractive index change. But the diffractive process is simulated by a stochastic component of the propagation direction. The propagating wave is substituted by one or several rays whose direction and relative power are modelled after the propagation probability. Very much like incident light decomposes from a continuous wave into a photon hit.

We made the necessary improvements in the manuscript to address the mentioned point.

23- Are every term and structure in the proposed design should be clearly and correctly presented not to mislead the reader.

The underlying physics is indeed rather simple. The chosen implementation is of course a new design and hence lacks descriptive terminology. But we made an effort to describe how it is anchored in existing approaches that are similar. We will make a re-labelling effort.

24- If it is possible to compare between the computational and experimental results.

Not yet on a full scale. We gave a few examples where this works for simpler models (such as mirrors or beam splitters) where the results are published and hence are used for verification of the method. There is of course not just the risk but the guarantee that the complex diffraction processes will lead to simulation errors. We list comparisons how – for smaller structures – these errors are introduced and how severe they are. For small structures that can be handled by other methods, these errors amount to some 3% of the total energy. For our much larger structures, these errors could – in the worst case – add up to 10-15% if all errors sum up destructively. If they sum up stochastically – a fair bit more likely – the expected errors are in the 7% range.

25-Finally, I recommend that the paper should be revised taking care of the above comments.

Thank you

Reviewer 3 Report

I have some serious concerns related to the presentation of the paper. The results are not well organized which creates confusion in several places. The English used in the paper is unacceptable as most of the sentences are not well structured. I suggest major revisions. I have the following suggestions:

 1) Reduce the over lengthy Abstract section. Only vital and obtained results should be provided in this section. There is no need to explain the history of the topic. 

2) In the introduction section, provide the recent studies related to the mitochondria lens and evaluate their performance (with proper references). 

3)  Figure 1 is not cited in the paper. And figure should be moved out of the introduction section. Please it in the section next to the Introduction section. 

4) Why figures on page 4 have no caption? And not cited in the paper.

5) In the table, for which wavelength the value of refractive indices are obtained?

6) Very poor sentence formation. I suggest revising the paper with the help of a native English speaker. 

7) Figure 2 is not cited in the paper. 

8) Figure 4 is cited after Figure 5. Keep the order in the paper, first figure 4 should be cited then figure 5. 

9) On page 10, just after figure 7, figure 10 is cited, why? 

10) What is the transmission loss of the waveguide? 

Author Response

1) Reduce the over lengthy Abstract section. Only vital and obtained results should be provided in this section. There is no need to explain the history of the topic.

We thank the reviewer for the valuable feedback. We made the necessary improvements in the abstract to address the mentioned point.

2) In the introduction section, provide the recent studies related to the mitochondria lens and evaluate their performance (with proper references).

We thank the reviewer for the valuable feedback. There are very limited publications on the lens mitochondria and their experimental performance is currently unknown. The properties as a light filter are inferred by the biology of the tree shrew’s eyes and the results of our simulation. There is currently, no alternative data that elaborates on the filter property. We do however provided references to show that the retinae do not suffer from UV damage and that hence a filter must be present in the system.

3)  Figure 1 is not cited in the paper. And figure should be moved out of the introduction section. Please it in the section next to the Introduction section.

Figure 1 is cited twice in the first paragraph in the introduction. Its serves as a base for our choice of model dimensions and as an illustration of the problem discussed in the paper.

We have moved it out from the introduction to the following section.

4) Why figures on page 4 have no caption? And not cited in the paper.

We thank the reviewer for the valuable feedback. We made the necessary improvements in the figure to address the mentioned point.

5) In the table, for which wavelength the value of refractive indices are obtained?

We thank the reviewer for the valuable feedback. The RIs are cited at the Fraunhofer d line (the centre line) at 587.56nm and the wavelength dependence of the RIs or Abbe numbers of the materials are not known. We made the necessary improvements in the table legend to address the mentioned point.

6) Very poor sentence formation. I suggest revising the paper with the help of a native English speaker.

We made the necessary improvements on the text.

7) Figure 2 is not cited in the paper.

We made the necessary improvements to address the mentioned point.

8) Figure 4 is cited after Figure 5. Keep the order in the paper, first figure 4 should be cited then figure 5.

We made the necessary improvements to address the mentioned point. Both figures are cited in the same sentence.

9) On page 10, just after figure 7, figure 10 is cited, why?

We thank the reviewer for the valuable feedback. We made the necessary amendment to address the mentioned point.

10) What is the transmission loss of the waveguide?

For the materials here, a few dB per meter. Since the structure is some 5m (5 micron) in size, attenuation and absorption can safely be assumed to be zero. The filter property originates exclusively from scattering and reflection. We have incorporated a new paragraph in the manuscript to provide further clarification on this point.

Round 2

Reviewer 2 Report

All comments have considered by authors

Reviewer 3 Report

I am willing to accept the paper in its current form.